# Bronchial Asthma, Airway Remodeling and Lung Fibrosis as Successive Steps of One Process

**DOI:** 10.3390/ijms242216042

**Published:** 2023-11-07

**Authors:** Innokenty A. Savin, Marina A. Zenkova, Aleksandra V. Sen’kova

**Affiliations:** Institute of Chemical Biology and Fundamental Medicine, Siberian Branch of the Russian Academy of Sciences, Lavrent’ev Ave 8, 630090 Novosibirsk, Russia; savin_ia@niboch.nsc.ru (I.A.S.); marzen@niboch.nsc.ru (M.A.Z.)

**Keywords:** asthma, airway remodeling, lung fibrosis, in vivo models, biomarkers

## Abstract

Bronchial asthma is a heterogeneous disease characterized by persistent respiratory system inflammation, airway hyperreactivity, and airflow obstruction. Airway remodeling, defined as changes in airway wall structure such as extensive epithelial damage, airway smooth muscle hypertrophy, collagen deposition, and subepithelial fibrosis, is a key feature of asthma. Lung fibrosis is a common occurrence in the pathogenesis of fatal and long-term asthma, and it is associated with disease severity and resistance to therapy. It can thus be regarded as an irreversible consequence of asthma-induced airway inflammation and remodeling. Asthma heterogeneity presents several diagnostic challenges, particularly in distinguishing between chronic asthma and other pulmonary diseases characterized by disruption of normal lung architecture and functions, such as chronic obstructive pulmonary disease. The search for instruments that can predict the development of irreversible structural changes in the lungs, such as chronic components of airway remodeling and fibrosis, is particularly difficult. To overcome these challenges, significant efforts are being directed toward the discovery and investigation of molecular characteristics and biomarkers capable of distinguishing between different types of asthma as well as between asthma and other pulmonary disorders with similar structural characteristics. The main features of bronchial asthma etiology, pathogenesis, and morphological characteristics as well as asthma-associated airway remodeling and lung fibrosis as successive stages of one process will be discussed in this review. The most common murine models and biomarkers of asthma progression and post-asthmatic fibrosis will also be covered. The molecular mechanisms and key cellular players of the asthmatic process described and systematized in this review are intended to help in the search for new molecular markers and promising therapeutic targets for asthma prediction and therapy.

## 1. Introduction

Asthma is the most common chronic inflammatory disease of the respiratory tract, characterized by leukocyte infiltration and tissue remodeling, with the latter generally referring to epithelial hyperplasia and collagen deposition [1]. Clinically, asthma is accompanied by airway inflammation, hyperresponsiveness, and airflow limitation, which can lead to respiratory symptoms such as coughing, wheezing, and shortness of breath [2]. The pathogenesis of asthma is complex and involves various genetic, environmental, and immunological factors [3,4].

One of the central features of asthma is airway remodeling, defined as changes in airway wall structure, including extensive epithelial damage, airway smooth muscle hypertrophy and hyperplasia, collagen deposition, subepithelial basement membrane thickening, and fibrosis [5]. Excessive proliferation of smooth muscle cells producing a wide range of pro-inflammatory and pro-fibrotic mediators may lead to amplified airflow obstruction and extracellular matrix (ECM) deposition, ultimately resulting in fibrosis in individuals affected by asthma [6,7]. Subepithelial fibrosis observed in asthma is associated with enhanced differentiation of bronchial fibroblasts into myofibroblasts—fibroblast-to-myofibroblast transition (FMT)—induced mainly by transforming growth factor-β (TGF-β) [8].

Fibrosis of alveolar structures is an important phenomenon, as it usually occurs in the pathogenesis of fatal and long-term asthma and may be associated with disease severity and resistance to therapy [9,10,11]. In asthma treatment, drugs targeting chronic inflammation and bronchodilators control asthma but have a negligible effect on the structural changes in the bronchi. Recent studies indicate that inflammation and remodeling of asthmatic bronchi can be driven independently [8,12]. Therefore, lung fibrosis can be considered a long-term and eventually irreversible consequence of asthma-induced airway inflammation and remodeling.

In the present review, morphological features and molecular mechanisms of asthma-associated airway remodeling and lung fibrosis, as well as the most common murine models and promising biomarkers of asthma progression and post-asthmatic fibrosis, will be discussed.

## 2. Bronchial Asthma as One of the Precursors of Lung Fibrosis: Etiology, Pathogenesis, and Morphological Characteristics

Bronchial asthma is a heterogenic disease characterized by persistent inflammation in the respiratory system, airway hyperreactivity, and reversible airflow obstruction, affecting approximately 300 million people worldwide [13,14]. Trends in asthma prevalence have fluctuated throughout the last decades; the overall number of asthma cases has remained consistent, though asthma-related deaths have decreased in recent years, reflecting improved therapeutic control. However, despite advances in modern healthcare, global asthma damage remains high, with about 450,000 asthma-related overall deaths and an economic burden that costs USD 50 billion annually [15,16]. In addition, it is one of the most widespread chronic lung pathology among pediatric patients [17].

### 2.1. Asthma Endotypes and Phenotypes

At the present time, asthma is considered an “umbrella” diagnosis, unifying several diseases with different clinical manifestations (phenotypes) and pathophysiological mechanisms (endotypes) [18]. According to the Global Initiative for Asthma (GINA) definition, “asthma phenotypes” are recognizable clusters of demographic, clinical, and/or pathophysiological characteristics [19], while the term “endotypes” describes a subset of asthma with distinct molecular mechanisms and treatment response [20]. Today, the best-researched type of asthma is eosinophilic asthma, the most common type of this disease, also called T2-high endotype [21]. The T2-high endotype includes the following phenotypes: early-onset atopic (responsive to steroids), late-onset non-atopic eosinophilic (refractory to steroids), and aspirin-exacerbated respiratory disease (surgical treatment, sensitive to leukotriene modifiers).

Early-onset atopic asthma is the archetypal asthma phenotype, with a well-defined early onset indicated by blood or sputum eosinophil count, serum IgE, high FeNO, and high total IgE, and is sensitive to inhaled corticosteroids (ICS) therapy. It is distinguished from T2-high non-atopic asthma by positive skin prick tests and increased IgE [22].

Late-onset eosinophilic asthma is a subset of T2-high asthma manifesting in adulthood, notable for its higher severity and steroid resistance. The majority of these patients also have comorbid chronic rhinosinusitis with nasal polyps. Generally, this phenotype is characterized by prominent blood and sputum eosinophilia refractory to ICS treatment and normal or slightly elevated serum IgE levels. Additionally, it is believed that inflammation in this phenotype is driven by the production of IL-5 and IL-13 by innate lymphoid cells. Some patients also have sputum neutrophilia, indicating that Th2/Th17 interactions are taking place [23].

Aspirin-exacerbated respiratory disease (AERD) is a subset of late-onset eosinophilic asthma, characterized by dysregulated arachidonic acid metabolism, cysteinyl leukotrienes production, elevated eosinophils in the blood and sputum, high severity from the onset, and frequent exacerbations. Aspirin is a potent cyclooxygenase inhibitor, and it shifts arachidonic acid metabolism from the cyclooxygenase to the 5-lipooxygenase pathway. This leads to the overproduction of cysteinyl leukotrienes, which are potent bronchoconstrictors responsible for the clinical symptoms and resistance to conventional therapy for AERD [24].

Non-eosinophilic asthma, also known as T2-low or non-T2, is a less understood endotype of asthma and is typically defined by the absence of T2-high asthma signs, such as eosinophilia and elevated IgE, the presence of neutrophilic or paucigranulocytic inflammation, and resistance to ICS. Mechanisms underlying the manifestation of T2-low asthma and the maintenance of neutrophilic inflammation are currently unknown, but they have been associated with chronic infection, obesity, smoking, and smooth muscle abnormalities [25]. Therapeutic options are quite limited and consist of tiotropium and macrolides [26].

### 2.2. Etiology and Pathogenesis of Bronchial Asthma

The etiology of bronchial asthma is currently unknown; however, there are plenty of risk factors, including genetic and environmental conditions [27]. Genetic factors include changes in the expression of several genes responsible for protein folding in the endoplasmic reticulum [27], epithelial [28], and eosinophil dysfunction [29]. Allergic airway diseases (such as allergic rhinitis) are also associated with an increased risk of asthma development [30]. Environmental factors include smoking (both active and passive) [31], air pollution (including automobile associated, such as black carbon and NO_2_) [32], obesity [33], and professional risk factors, such as flour dust, animal and plant enzymes, tree resins, tobacco, polyisocyanate, acids, anhydrides, and metals [34,35].

Allergic asthma is considered one of the most widespread asthma types, developing due to sensitization to environmental allergens, mostly house dust, plant pollen, and mushroom spores [36]. After sensitization, asthma symptoms usually develop during second contact with the allergen [37]. Allergic reactions, activating the IgE dependent pathways, are the most common mechanism underlying asthma.

IgE is the main effector of type 1 hypersensitivity, underlying the development of asthmatic inflammation [38]. Its synthesis occurs either by direct class-switch recombination from IgM in germinal centers or through a “sequential” switch from IgM to IgG1 and then from IgG1 to IgE outside of germinal centers. The high-affinity receptor of IgE (FcεRI) is expressed on mast cells and basophils as a tetramer and on monocytes and dendritic cells as a trimer.

During the sensitization step of asthma development, IgE focuses the allergen on the cell surface through FcεRI, leading to the procession of the antigen-IgE complex and presentation through the major histocompatibility complex class II molecules, lowering the threshold for T-cell activation during the allergen challenge [39]. During the next contact with the allergen, inflammation is initiated when the antigen contacts IgE, presenting on all mast cells and basophiles. After contact, cells degranulate, releasing such mediators as histamine, heparin, proteases, and pro-inflammatory cytokines, which are responsible for the chemotaxis of inflammatory cells.

In addition to classic IgE, there is a cytokinergic IgE that facilitates asthmatic inflammation in the absence of allergens, making allergen avoidance an ineffective therapeutic strategy [40].

CD4+ lymphocytes also take part in the development of allergic asthma. After contact with antigen, T helpers type 2 (Th2) secrete pro-inflammatory cytokines, such as IL-4, IL-5, IL-9, and IL-13, which stimulate IgE production and inflammatory cell migration [41,42,43,44,45]. In turn, T helpers type 1 (Th1) start to secrete IL-2 and IFN-γ, activating macrophages and enhancing the cell immune response. T-cell immune response is additionally controlled by IL-1, IL-4, IL-12, and IL-18, secreted by dendritic cells [46]. The cascade of the aforementioned reactions leads to persisting inflammation in the lungs.

About one-third of bronchial asthma patients are believed to have non-allergic asthma, mediated by non-Th2 cytokines, including IL-17 and TNF-α, and characterized by the absence of allergen reactions in the skin prick tests and a decreased or unaffected amount of IgE, contrary to allergic asthma [47,48]. The mechanisms of non-allergic asthma development are currently unknown, but it is supposed that there are two parts to its pathogenesis: dysregulation of the neutrophilic immune response due to lung inflammation [49], and defects in IL-17 mediated signaling pathway [50], leading to the persisting inflammation [51].

Another group of cells that play a significant role in asthma and post-asthmatic fibrosis development are innate lymphoid cells (ILCs). It is a group of loosely related lymphocytes, characterized into five subgroups based on functions, origins, transcription factors, and cytokine expression patterns: natural killer (NK) cells, ILC1s, ILC2s, ILC3s, and lymphoid tissue-inducer cells [52]. They are abundantly present in the tissue of organs performing barrier functions such as intestines, lungs, and skin. For quite some time, ILC2s have been established as crucial mediators of lung allergy, airway inflammation, and fibrosis, thus affecting the pathogenesis and clinical course of many respiratory diseases, like, for instance, asthma, cystic fibrosis, and chronic rhinosinusitis [53]. More specifically, ILC2s are activated by the alarmin cytokines IL-22 and IL-33, produced by the lung epithelium after contact with allergens, infections, and other injurious stimuli. After activation, ILC2s start producing IL-5, IL-13, and amphiregulin, which in turn recruit and stimulate eosinophils to release profibrotic cytokines such as TGF-β, PDGF, and IL-13, promoting the fibroblast-to-myofibroblast transition [54].

### 2.3. Pathomorphological Changes in the Lungs during Asthma Development

Pathomorphological changes in the bronchial asthma lungs can be divided into two patterns: alterations in bronchial epithelium and smooth muscles, prominent signs of asthma exacerbations, and subepithelial fibrosis, a characteristic of long-term asthma [55]. All these pathological changes lead to bronchial obstruction, which is reversible at the early stages of the disease and irreversible at the later ones.

During acute asthma development, hyperplasia and metaplasia of the goblet and epithelial cells of the bronchial epithelium, leading to mucus hyperproduction, thickening of the airways, and bronchial obstruction, are observed [56]. Moreover, in severe asthma exacerbations, large and small airways are often obstructed by mucus plugs with an admixture of inflammatory cells (mostly eosinophils in the case of allergic asthma) [57]. An additional factor leading to the formation of mucus plugs is the dysfunction of ciliated cells due to airway inflammation, characterized by a decrease in the frequency of its fluctuations as well as dyskinesia and disorientation of the cilia [58].

Spasm of the bronchial smooth musculature—bronchoconstriction—is another factor leading to airway obstruction. Under physiological conditions, bronchial smooth muscles provide mechanical stability to the airways without cartilage. However, hyperreactivity of asthmatic airways decreases smooth muscle sensitivity threshold, following spasm and reversible airway obstruction [59]. The accumulation of smooth muscle cells due to their hypertrophy and hyperplasia is another component of asthma pathomorphological changes, leading to airway thickening [60]. Moreover, it is believed that smooth muscle cells may support airway remodeling through the secretion of pro-inflammatory mediators, matrix and cell adhesion proteins, and other stimulatory molecules, affecting the further migration and activity of inflammatory cells [61].

The major characteristic of chronic asthmatic inflammation is subepithelial airway and, in some cases, lung fibrosis, consisting of connective tissue growth in the basal membrane and submucosal area. However, changes in the airways, leading to lung fibrosis, are present even in the earliest stages of asthma [55,62].

## 3. Morphological Characteristics and Molecular Mechanisms of Asthma-Associated Airway Remodeling and Lung Fibrosis

The exaggerated chronic inflammation typical of chronic pulmonary diseases, including lung cancer, interstitial lung diseases, asthma, chronic obstructive pulmonary disease, and other muco-obstructive lung diseases, such as cystic fibrosis and non-cystic fibrosis bronchiectasis, can induce molecular reprogramming with subsequent self-sustaining aberrant and excessive pro-fibrotic tissue repair [63]. Persistent lung inflammation as a result of a long course of bronchial asthma is one of the factors leading to airway remodeling and fibrosis development [16]. Airway remodeling is present in all asthma phenotypes independently of disease severity, and the extent of structural changes due to airway remodeling remains unchanged independently of symptoms control or medication use [64]. Lung fibrosis occurs generally in fatal asthma, and fibrotic changes are associated with disease severity and resistance to therapy [9,65].

### 3.1. Airway Remodeling

Structural changes in the airways associated with the progression and chronization of asthma and other chronic inflammatory diseases of the lungs are commonly referred to as “airway remodeling”, characterized by cellular and extracellular changes in large and small airways. These changes consist of a decrease in epithelial barrier integrity leading to goblet cell hyperplasia and mucus hypersecretion [66,67,68], smooth muscle cell proliferation [69,70], increased angiogenesis [71,72], and fibroblast/myofibroblast accumulation with deposition of ECM components in the lung tissue resulting in subepithelial fibrosis [73,74,75] (Figure 1).

Epithelial alterations in asthmatic lungs include breakdown in epithelial tight junction integrity, shedding of the epithelium, loss of ciliated cells, and goblet cell hyperplasia [71]. Mucus hypersecretion is usually found in large and medium airways in asthmatic patients. MUC5AC and MUC5B were identified as key mucins in mucus hypersecretion in asthma; their levels increased around the airways of asthmatic patients and in experimental models of chronic asthma in mice [76]. A key driver in increased production of mucus is goblet cell hyperplasia, found in mild/severe asthma and regulated by Th2 cytokines (IL-4, IL-5, and IL-13) as well as IL-1b, TNF-a, COX-2, and their associated intracellular signaling pathways [77]. Overall, goblet cell hyperplasia and excessive mucus production can lead to mucus plugging in the airways and subsequent airway obstruction.

Airway smooth muscle (ASM) cells constitute the main structural cells within the bronchi, and the remodeling of ASM cells, represented by their proliferation (hyperplasia) and increased cell size (hypertrophy), is considered to be the primary cause of airway obstruction [71]. Moreover, ASM cells participate in the inflammatory and airway remodeling processes through the expression of integrins, cellular adhesion molecules (CAMs), pro-inflammatory cytokines (TNF-a, IL-1b), and chemokines (RANTES, eotaxin, and IL-8) [65,78].

Angiogenesis is the process of new blood vessel formation from preexisting endothelial-lined vessels. An abnormal increase in the number and size of microvessels within bronchial tissue in asthmatic airways is observed mainly in the bronchial smooth muscle layer as well as through the capillary network in the lamina propria [71,79]. Hypoxia-inducible factor (HIF) and vascular endothelial growth factor (VEGF), key players in angiogenesis, increase the permeability of these abnormal blood vessels, resulting in vessel dilation and edema, which contribute to airway narrowing [79]. In addition, remodeled vessels in the airways of patients with asthma may promote the extravasation of inflammatory cells, the release of plasma-derived inflammatory mediators and cytokines, and abnormal cell growth and proliferation leading to asthma pathology [72].

### 3.2. Subepitelial Fibrosis as Irreversible Component of Airway Remodeling

Chronic inflammatory airway diseases such as chronic severe asthma and chronic obstructive pulmonary disease (COPD) lead to bronchial subepithelial fibrosis, fixed airway obstruction, and in some cases irreversible structural changes in the respiratory tract and lung tissue [80]. Fibrosis is the major contributor to the pathology of chronic respiratory diseases, and the accumulation of fibrotic tissue is associated with more severe disease and a potential loss of sensitivity to therapy [81]. In the case of bronchial asthma, lung fibrosis develops after recurrent asthma attacks and leads to a progressive decline in lung function [82].

As a key component of asthmatic remodeling, subepithelial fibrosis, a distinct type of asthmatic lesion, involves the deposition of ECM proteins such as collagen types I, III, and V, fibronectin, hyaluronan, laminin α2/β2, tenascin, periostin, versican, decorin, lumican, and various proteoglycans within the lamina reticularis of the airways, resulting in thickening of the basement membrane [72,83]. Increased deposition of ECM components, including fragmented and disorganized fibrillar collagen, has also been demonstrated in the lamina propria of large and small airways in patients with asthma [74]. The degree of subepithelial fibrosis is often linked to asthma severity; the amount of collagen in the airways is usually higher in patients with moderate and severe asthma compared to patients with mild disease, and the degree of subepithelial fibrosis is inversely correlated with forced expiratory volume in the first second (FEV1), indicative of pulmonary functions [84].

Fibroblasts, the main connective tissue cells, are large, flat stellate cells that reside in close proximity to the basal epithelium [71]. In an inflammatory environment such as asthmatic airways, fibroblasts are activated or differentiated into myofibroblasts, which secrete pro-inflammatory mediators and ECM proteins, leading to the accumulation of collagen fibers around large and small bronchi, and the degree of fibrosis correlates with an increased number of fibroblasts and myofibroblasts in asthmatic lungs [85,86]. Persistent inflammation in asthmatic airways leads to a decrease in epithelial barrier integrity, stimulating the production of ECM components by airway epithelial cells and smooth muscle cells, which in turn stimulate the production of collagen, fibronectin, and other ECM components by lung fibroblasts and myofibroblasts [67]. Additionally, products of ECM degradation, referred to as “matrikines”, modulate the production of ECM components, forming a closed positive-feedback loop of asthmatic airway remodeling [87]. So, the main cause of subepithelial fibrosis is the imbalance of ECM synthesis and degradation, leading to excessive scarring and reducing compliance and dilator responsiveness in fibrotic airways [88]. In total, these processes lead to the thickening of asthmatic airway walls, occlusion, and later complete obliteration of large and small airways [73].

Basement membrane thickening and the formation of fibrotic foci below the basement membrane due to excessive deposition of ECM proteins is an early and universal feature of airway wall remodeling in asthma [89]. The standard detection of structural alterations in asthmatic lungs with fibrosis is through direct histological analyses of airway tissues obtained post mortem, surgically, or by flexible bronchoscopy [90]. The noninvasive assessment of lung structure using both computed tomography (CT) and ultrashort or zero echo time magnetic resonance imaging (MRI) techniques remains the gold standard for structural lung imaging in many clinical indications, including bronchial asthma and COPD [91]. Hyperpolarized (HP) gas MRI with inhaled ^3^He and ^129^Xe, a novel method for functional and microstructural imaging of the lungs, has great potential as a clinical tool for early detection and improved understanding of pathophysiology in patients with a wide range of pulmonary disorders and is increasingly of interest today for both clinicians and scientists [91].

### 3.3. Molecular Mechanisms of Asthma-Associated Lung Fibrosis

As described above, airway remodeling in asthma consists of subepithelial fibrosis, deposition of ECM components, goblet cell hyperplasia and mucus overproduction, proliferation of smooth muscle cells, and disrupted integrity of the airway epithelial barrier. Several pro-inflammatory pathways regulate all these features and are thought to contribute to asthma development. These pathways include TGF-β, STAT-3, and NF-κB pathways, the peroxisome proliferator-activated receptors (PPARs) pathway, the protease-activated receptor-2 (PAR-2) pathway, and the fibroblast-to-myofibroblast transition (FMT).

#### 3.3.1. TGF-β

Transforming growth factor β (TGF-β) superfamily of ligands are multifunctional regulators involved in various biological processes in the lungs, such as alveolarization, epithelial barrier functioning, cell differentiation, and proliferation [92]. In normal airways, TGF-β exerts its anti-apoptotic effect through the Smad2/3 pathway during normal healing processes. However, in asthmatic airways, TGF-β induces a pro-apoptotic effect in airway epithelial cells. When airway epithelial cells are under continuous stress exposure to allergens, the p38 mitogen-activated protein kinase (MAPK) pathway is activated, and TGF-β initiates apoptosis, which leads to the loss of epithelial barrier integrity [93]. Additionally, TGF-β plays a role in the development of subepithelial fibrosis, promoting differentiation of fibroblasts into myofibroblasts and stimulating the release of connective tissue growth factor (CTGF), which enhances the adhesion and migration of mesenchymal cells [94].

#### 3.3.2. STAT-3

The signal transducer and activator of transcription (STAT) family consists of seven transcription factors participating in cell activation. There is evidence that STAT-6 takes part in the initiation of Th2-mediated lung inflammation in bronchial asthma, with the IL-4/IL-13/STAT-6 pathway being a key modulator of asthmatic inflammation [95]. However, the involvement of another member of the STAT family, STAT-3, in the development of bronchial asthma is still unclear. Recently, it has been shown that STAT-3 is crucial for the polarization of Th17 cells, taking part in the neutrophil-mediated inflammation in bronchial asthma and in the Th2-mediated immune response in general. Additionally, STAT-3 is involved in the polarization of alternatively activated M2 macrophages and fibroblasts, leading to the elevated production of ECM components and the development of fibrosis [96].

#### 3.3.3. NF-κB

Nuclear factor kappa-light-chain enhancer of activated B cells (NF-κB) is a transcription factor composed of several regulatory molecules activated by TNF-α and IL-1β cytokines, controlling the expression of a multitude of inflammatory genes and biological effects such as proliferation, differentiation, apoptosis, and tissue remodeling [97]. Several studies have demonstrated that NF-κB is activated in the ovalbumin (OVA) allergic asthma model predominantly in airway epithelial cells, together with enhanced expression of MIP-2 and eotaxin mRNAs—NF-κB-regulated chemokines [98]. Consequently, there have been several attempts at regulating the NF-κB pathway as a potential approach for asthma treatment [99].

#### 3.3.4. Peroxisome Proliferator-Activated Receptors (PPARs)

Peroxisome proliferator-activated receptors (PPARs) are nuclear hormone receptors consisting of three subunits: PPARα, PPARβ/δ, and PPARγ. They were initially recognized as regulators of lipid and glucose metabolism [100]. They also play a certain role in the regulation of biological processes such as differentiation, proliferation, survival, apoptosis, motility, inflammation, and immune response. Immune cells in the inflamed airways, such as dendritic cells, eosinophils, macrophages, mast cells, monocytes, and neutrophils, have been found to express PPARs [101]. In general, the expression and activities of PPARs are associated with protection against asthma or a reduction in asthma severity, whereas impairment of PPAR functions and expression exacerbates the disease. For example, PPARα and PPARγ down-regulate the expression of matrix metalloproteinases and, as a result, ECM degradation, and PPARβ/δ and PPARγ suppress the proliferation of lung fibroblasts and their differentiation into myofibroblasts, reducing the overall collagen production and ECM component deposition. PPARγ also inhibits epithelial and smooth muscle cell hyperplasia and blocks mucus overproduction [102].

#### 3.3.5. Protease-Activated Receptor-2 (PAR-2)

Protease-activated receptors (PARs) are a family of G-protein-coupled receptors with four members, PAR-1, PAR-2, PAR-3, and PAR-4, activated by serine proteases secreted by inflammatory cells or microorganisms [103]. Among PARs, PAR-2 has a wide expression pattern and has been linked to allergic airway inflammation [104]. PAR-2-mediated activation of airway epithelial cells has been reported to release a number of factors and inflammatory mediators, including metalloproteinases, IL-8, IL-6, GM-CSF, and various chemokines such as eotaxin and CCL-2, that play an important role in asthma pathogenesis via polarizing the immune response toward the Th2 phenotype and attracting innate and adaptive immune cells to the airways [103].

#### 3.3.6. Fibroblast-to-Myofibroblast Transition (FMT)

Fibroblast-to-myofibroblast transition (FMT) is a phenomenon that occurs both under physiological and pathological conditions. A vast amount of myofibroblasts in connective tissue due to increased transition from fibroblasts and disrupted apoptosis is related to the pathologic processes of wound healing and chronic inflammation. As for bronchial asthma, FMT has been reported in airway remodeling and subepithelial fibrosis development [105]. FMT consists of two main steps: fibroblasts develop a translational phenotype, known as proto-myofibroblasts, and then differentiate into mature myofibroblasts. FMT is facilitated by mechanical tension in the altered tissue and is accompanied by the secretion of several cytokines [106]. In asthma, there are two groups of factors stimulating FMT: humoral agents, such as growth factors, cytokines, and chemokines, and mechanical factors, including intercellular and cell–ECM interactions. Due to the complicated pathogenesis of asthma and a wide variety of endo- and phenotypes, many FMT stimuli may interact with each other, leading to FMT induction [107].

## 4. Murine Models of Asthma and Asthma-Associated Lung Fibrosis

Murine models are frequently used in all fields of biological research due to their relatively close similarity to humans as well as their efficient and simple breeding and housing. In the last few years, the number and diversity of available mouse models of different human pathologies have increased exponentially. Asthma is a complex, multifactorial disease that requires a thorough investigation in the context of a whole organism. Although in vitro and in silico methods help elucidate certain mechanistic pathways, mouse models of asthma remain the most physiological replication of the different parts of asthma pathogenesis [108]. Despite differences in anatomical structures as well as cellular and functional distinctions between mouse and human lungs, mouse models are indispensable tools in the investigation of asthma and other complex diseases.

Today, the majority of in vivo murine studies of asthma use one of the two models: either the asthma model induced by chicken egg white ovalbumin (OVA) or by extracts of different allergens, the most common being the extract of household dust mite (HDM) (Table 1).

### 4.1. Ovalbumin (OVA)-Induced Asthma

Among mouse models of acute and chronic asthma, asthma induced by ovalbumin (OVA), the primary component of eggs’ white, is the most common model of asthma-like symptoms in mice [109] (Table 1). The classical model of acute asthma is induced by sensitization to OVA through intraperitoneal injection of an OVA/aluminum hydroxide mixture, followed by cyclic inhalations of OVA aerosol [109,110]. Since OVA lacks inherent allergenic activity, during the sensitization step of the induction, it is usually injected together with an adjuvant compound to boost allergenic reactions in the airways, the most common being aluminum hydroxide and potassium aluminum sulfate, although adjuvant-less protocols also exist. During sensitization, airway dendritic cells encounter allergens, process them, and present them to CD4+ T-cells, which, in turn, switch naïve B-cells to OVA-specific B-cells. During cyclic inhalation of OVA, sensitized mast cells and basophils undergo degranulation, leading to the release of mediators, chemokines, and cytokines, hypersecretion of mucus, alteration of airway epithelial cells, and airway inflammation. This model reflects the earliest stages of asthma development, primarily inflammation of the respiratory tract [111]. A further increase in the number of OVA inhalation cycles leads to the development of chronic inflammatory changes in the airways, characterized by the reduction of inflammation and the emergence of reliable signs of airway remodeling as well as fibrotic changes in the lung tissue [14,119] (Table 1).

The advantages of the OVA-induced asthma model are the wide availability of inducing agents, robustness, and high reproducibility. The main disadvantage of this model is long-standing concerns regarding the clinical relevance of OVA as an asthma etiological factor in translating scientific findings from the murine models to patients.

This model has undergone several variations in an attempt to diversify it. For example, to stimulate the immune response, adjuvant injections were replaced by NO_2_ in order to more accurately reveal the role of air pollutants in asthma development [120]. To support an association between helminth infection and allergic/autoimmune disorders, co-administration of somatic antigens of Echinococcus granulosus simultaneously with OVA administration was performed, and intensification of allergic airway inflammation was detected in this case [121].

Doubts regarding OVA relevancy in human asthma development pushed investigators to try another asthma inducing antigens, such as house dust mite [122] and aspergillus [123] extracts, which are more common allergens in humans. Another challenge of this model is chronic antigen influence. In mice, inflammatory changes are relatively acute, while in humans, asthma is a long-term, chronic disease, exhibiting airway remodeling signs in the late stages [124]. Unfortunately, long-term antigen exposure in some mouse lines, such as Balb/C, led to tolerance development and a decrease in inflammation and airway hyperreactivity [125], pointing limitations of this model.

### 4.2. House Dust Mite (HDM) and Other Allergenic Extract-Induced Asthma

House dust mites (HDMs) are small but extremely complex organisms, ranging from 20 to 50 μm in size and living in humid environments of human habitation. These microscopic organisms are widely recognized as a primary source of indoor allergens, leading to the emergence of allergic diseases, including allergic bronchial asthma [126], with approximately 50% of individuals with asthma having an allergic response to HDMs [127]. HDM sensitization is usually mediated by aerosolized mite feces or fragmented bodies present in household dust. Additionally, household dust contains a large spectrum of environmental factors, such as bacteria and fungi, which, in association with HDM allergens, elicit activation of the immune system, thus being adjuvants to HDM sensitization.

Out of approximately 30 groups of HDM allergens, only several exhibit proteolytic activity and thus are the most clinically relevant groups. After inhalation, HDM allergens react with airway epithelial and immune cells, leading to the degradation of epithelial barrier integrity due to their proteolytic activity. HDM allergens stimulate protease-activated receptors in the bronchial epithelium, resulting in the release of adenosine triphosphate (ATP), which is considered a damage-associated molecular pattern (DAMP), activating ADAM10 and mediating reactive oxygen species (ROS) production, further damaging the epithelial barrier. Furthermore, HDMs interaction with the airway epithelium leads to the activation of Toll-like receptors (TLRs) and NOD-like receptors (NLRs), eliciting an innate immune response and the development of an allergic reaction [128].

The advantages of this model are the presence of airway hyperresponsiveness (AHR), one of the defining characteristics of asthma, and its high clinical relevance compared to the OVA-induced asthma model [129]. The biggest disadvantage is the high variability of HDM extracts, differing in composition and mice’s airway responses even in the same lot from one manufacturer [130].

There are a lot of variations of this model in mice using extracts of different allergens, such as cockroach extract [131], common ragweed extract [132], latex extract [133] (as of today seldom used as an asthmatic model) and extracts from several Aspergillus species, mainly fumigatus [134]. Additionally, a combination of HDMs, ragweed, and Aspergillus fumigatus extracts called DRA is also used to induce chronic asthma in mice, closely mimicking human asthma. The additional advantage of this model is that it overcomes allergen tolerance through repeated exposures to the allergens in mice [117].

## 5. Molecular Markers of Asthma Progression and Post-Asthmatic Fibrosis Development

Successful therapy and management of asthma require the identification of key markers of asthma progression and chronization. Today, asthma clinical symptoms, such as progressive dyspnea, chest tightening, coughing, and wheezing, as well as symptoms of fatal asthma—previous episodes of severe asthma exacerbations, admission to the intensive care unit (ICU), and an uncontrollable course of the asthma—are considered signs of an unfavorable course of the disease [135]. Despite multiple studies, molecular markers of asthma progression, uncoupled from clinical symptoms, are still in the early stages of research. Currently, biomarkers related to Th2-high (allergic) asthma or Th2-low (non-allergic) asthma are clearly established and researched separately.

Th2-high-related biomarkers are represented by sputum eosinophils, blood total eosinophil count, serum IgE, fraction of exhaled nitric oxide (FeNO) in breath condensate, serum periostin, and levels of IL-4, IL-5, and IL-13 in sputum and broncho-alveolar lavage fluid (BALF) [136,137]. Another biomarker of Th2-high asthma is eosinophilic cationic protein (ECP), one of the major cationic granule proteins released by activated eosinophils [138]. Serum and sputum concentrations of ECP were shown to correlate with asthma severity, sputum eosinophils, and FEV1 in treated asthmatics and are most useful when distinguishing between mild and severe asthmatics [139].

Lipoxins are endogenously produced eicosanoids with potent anti-inflammatory properties, playing an important role in chemotaxis and related signal transduction [140]. It was found that the expression of lipoxin A4 is decreased in the airways and peripheral blood of patients with severe asthma when compared to mild cases due to persistent activation of lymphoid cells and eosinophils [141]. The reasons for lipoxin concentration decreases in severe asthma patients are not clear yet, but it is theorized that they could be related to oxidative stress. Thus, increasing the generation of lipoxin A4 in the airways of asthmatic patients could constitute a new therapeutic approach for patients with severe asthma [142].

Th2-low asthma is currently less understood than Th2-high asthma, leading to fewer biomarkers overall. Th2-low asthma biomarkers include sputum neutrophils, levels of IL-17, TNF-α, IFN-γ, and IL-6 in sputum, BALF, and bronchial biopsies [136]. Additionally, expression of airway mucosal CCL26 seems to be one of the best biomarkers for differentiating between Th2-high and Th2-low asthma [143].

The latest available studies present miRNAs—small non-coding RNAs regulating gene expression at the post-transcriptional level by binding to the target mRNA—as a useful tool for predicting the effectiveness of therapy, early diagnosis of exacerbations, and assessing patient compliance for different groups of drugs used in asthma [144,145]. Several research studies have highlighted a correlation between the severity of asthma responses and the concentration of individual miRNA molecules in the blood and their direct impact on biological processes. miRNA-155-5p and miRNA-532-5p levels in blood serum were shown to correlate with the response of asthma patients to inhaled glucocorticosteroids [144]. Serum miRNA-21 can be successfully used as a non-invasive diagnostic marker of asthma, showing a positive correlation with blood and sputum eosinophilia and IL-4 concentration [146]. As it has been reported, increased levels of miRNA-146a in the airway epithelial cells and serum of adult patients indicate necessitating higher doses of inhaled steroids [147]. miRNA-144-3p levels were increased in airway biopsies and serum from severe asthmatics and associated with higher doses of corticosteroids. Moreover, its presence correlates directly with blood eosinophilia and overexpression of genes involved in asthma pathogenesis [148].

Due to similar clinical symptoms, there are certain challenges in differential diagnostics between asthma and chronic obstructive pulmonary disease (COPD), not to mention the asthma-COPD overlap (ACO), a condition characterized by persistent airflow limitations together with several distinguishing features of asthma. ACO is associated with a rapid decline in lung function, frequent exacerbations, and higher mortality when compared to either asthma or COPD alone [149]. Thus, there is an urgent need for molecular biomarkers capable of distinguishing between asthma, COPD, and ACO at the earlier stages, when clinical symptoms have not yet manifested. Bronchial biopsy is the gold standard of airway remodeling diagnostics; however, it is an invasive procedure and not all patients can undergo it, driving considerable efforts to identify potential biomarkers for long-term structural changes of the airways in asthma patients. However, the data we found regarding biomarkers of asthma-associated airway remodeling and potential post-asthmatic fibrosis are very sparse. To the best of our knowledge, below is the most comprehensive summary:IL-8 (CXCL8) is an anti-inflammatory chemokine produced by a wide range of cell populations, such as leukocytes, epithelial, and endothelial cells, together with fibroblasts. It takes part in many human diseases, such as atherosclerosis, inflammatory bowel disease, sepsis, acute lung injury, and asthma [150]. Vascular endothelial growth factor A (VEGFA) is a member of the VEGF protein family. It is primarily a major regulator of both physiological and pathological angiogenesis. However, VEGFA has many additional functions, including monocyte chemoattraction, osteoclast-mediated bone formation, and neuronal protection [151]. In asthma, it has been reported that IL-8 and VEGFA could be used in tandem as biomarkers to distinguish between asthma, COPD, and ACO, since IL-8 was highly sensitive, while VEGFA was highly specific to the difference between ACO and non-ACO patients [152];YKL-40, also known as the human cartilage glycoprotein-39, is a chitinase-like enzyme detectable in serum and airways. It plays a diverse role in cell proliferation, differentiation, survival, inflammation, and tissue remodeling [153,154]. Expression levels of YKL-40 were found to correlate with the probability of severe asthma and irreversible airway obstruction in asthmatic patients [155,156]. Moreover, it was reported that YKL-40 expression increases during asthma exacerbations and could predict a decline in lung function [157];Tissue inhibitor of metalloproteinases 1 (TIMP-1) is a protein with multiple functions, with the primary being the preservation of tissue integrity through controlling matrix metalloproteinases. Its other functions include, but are not limited to, the regulation of wound healing [158], regulation of cell proliferation, and signal transduction [159]. In asthma pathogenesis, TIMP-1 enhances eosinophilic inflammation and promotes macrophage polarization toward the M2 phenotype in the airways. It was also found that high levels of serum TIMP-1 were negatively correlated with FEV1 values in patients with severe asthma [160];Neutrophil gelatinase-associated lipocalin (NGAL), also known as oncogene 24p3 or lipocalin 2, is a member of the lipocalin family involved in the regulation of cell division, differentiation, cell-to-cell adhesion, and survival. Its small size, secreted nature, and relative stability led to its investigation as a diagnostic and prognostic biomarker in numerous diseases [161]. NGAL also functions as an innate antibacterial factor in sputum. There are several reports that expression levels of NGAL can be a distinguishing marker between asthma, COPD, and ACO [162,163];Hemopexin is a plasma glycoprotein with one of the highest binding affinities to heme, functioning as a heme scavenger and a second line of defense against hemoglobin-mediated oxidative damage during intravascular hemolysis, after haptoglobin. Moreover, it is an acute phase protein, expressing predominately in the liver, with its synthesis increasing mainly during inflammation [164]. Additionally, the role of hemopexin in the central and peripheral nervous systems is currently being investigated [165]. A recent report has shown that among hemoplexin, ceruloplasmin, and haptoglobin levels in the serum, hemopexin was the best-performing biomarker in differentiating between COPD and asthmatic patients [166];Ncf1 protein, also known as p47phox, is a principal component of the NADPH oxidase 2 complex, which mediates the induction of ROS in response to inflammatory stimuli. It is confirmed, that Ncf1 is associated with a variety of chronic inflammatory diseases both in animals and humans, such as rheumatoid arthritis and systemic lupus erythematosus [167]. Regarding the role of Ncf1 in lung diseases, it was reported that patients with primary non-small cell lung cancer achieve longer progression-free survival with higher levels of Ncf1 [168]. In asthmatic diseases, Ncf1 regulates the development of allergic inflammation through the induction of T-regulatory cells and control of T-cell-mediated inflammation, while the deficit of the Ncf1 gene ameliorates asthma development in the mice and is correlated with asthma severity, which points to the possible role of Ncf1 as a prognostic asthma biomarker [169].

Thus, a number of diverse molecules that can help in the differentiation of chronic lung pathologies, accompanied by structural changes in the airways as well as lung fibrosis, exist today. However, the data presented above clearly demonstrate that there is still an urgent need for more sensitive and specific biomarkers, since the present-day biomarkers are relatively sparse and do not cover all of the diagnostic and prognostic challenges associated with these diseases.

## 6. Conclusions

As of today, our understanding of asthma and asthma-associated pathologies is constantly evolving. However, due to the heterogeneity of asthma, in the foreseeable future there will always be a need for more specific and sensitive biomarkers, allowing physicians to not only diagnose asthma correctly but also stratify the risks of asthma exacerbations and long-term effects, including but not limited to such consequences of asthma as airway remodeling and lung fibrosis. There is hope that future studies of pathological features and key points of molecular mechanisms will allow us to develop personalized medicine approaches for asthma diagnoses and treatment. Overview and systematization of modern knowledge concerning the main stages of asthma development and progression, starting from airway inflammation and ending with irreversible structural changes and fibrotic transformation of the respiratory tract, and in some cases, lung parenchyma, as well as underlying molecular mechanisms, can be implemented in the search for new molecular markers and promising therapeutic targets of asthma and post-asthmatic fibrosis.

## Figures and Tables

**Figure 1 ijms-24-16042-f001:**
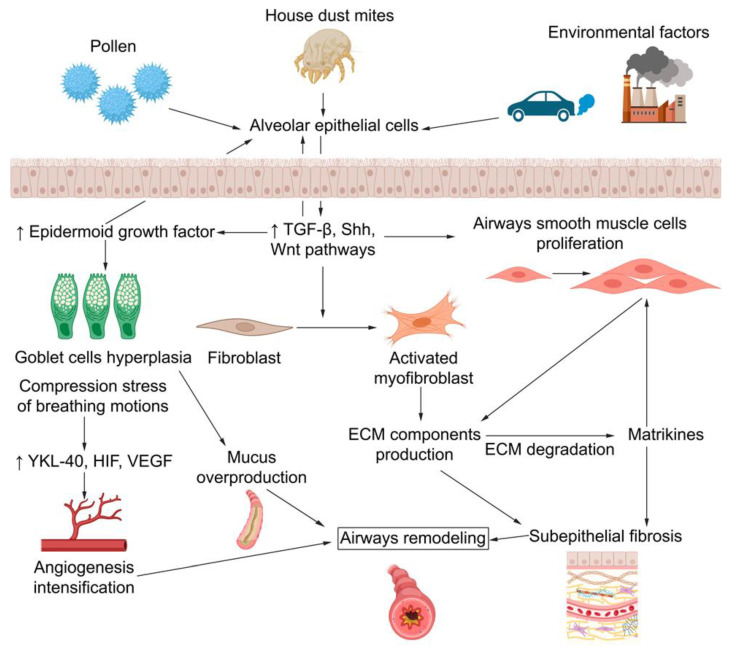
Principal pathophysiological components of airway remodeling emergence in allergic asthma. After contact with allergens, alveolar epithelial cells (AECs) initiate several processes through TGF-β, Sonic hedgehog (Shh), and Wnt pathways, such as airway smooth muscle cell proliferation, goblet cell hyperplasia, and fibroblast-to-myofibroblast transition. Together, these biological processes lead to the production and deposition of extracellular matrix (ECM) components, which, alongside ECM degradation products, lead to the emergence of subepithelial fibrosis and airway remodeling.

**Table 1 ijms-24-16042-t001:** Overview of murine models of asthma.

Model	Histological Characteristics	Modeling Object	Advantages and Disadvantages	References
Ovalbumin (OVA)-induced asthma Inducing agent: Sensitization–OVA/aluminum hydroxide intraperitoneal injection; Challenge–OVA inhalation (short duration in acute asthma, long duration in chronic asthma)	Acute asthma: ✓ Peribronchial inflammatory infiltration ✓ Goblet cells proliferation and increased mucus secretion Chronic asthma: ✓ Decline of inflammatory infiltration intensity compared to acute asthma ✓ Proliferation of smooth muscle cells ✓ Peribronchial and perivascular subepithelial fibrosis	Acute asthma: Airways inflammation, typical for early stages and exacerbations of asthma Chronic asthma: Irreversible changes in the lungs, typical for long-term, chronic asthma	Acute asthma: +Simplicity +Reproducibility +Short experiment duration −No airways remodeling −High variability of other asthma parameters (airways hypersensitivity/hyperreactivity, inflammation intensity) Chronic asthma: +Reflection of some key asthma characteristics in humans (goblet cells hyperplasia/metaplasia, local smooth muscle cells proliferation; peribronchial and perivascular fibrosis) −Long induction time −Relevancy of inducing agent regarding human asthma	Acute asthma: [109,110,111,112,113] Chronic asthma: [14,114]
Household dust mite (HDM)-induced asthma Inducing agent: Sensitization–HDM extract intratracheal instillation; Challenge–HDM extract intranasal instillation	✓ Eosinophilic and neutrophilic infiltration of airways and lung tissue ✓ Goblet cells proliferation and increased mucus secretion	Airways inflammation characterizing mild asthma	+Clinical relevancy +Simplicity +Presence of airways hyperresponsiveness −Dependence on activity and concentration of HDM extract leading to reproducibility issues	[114,115,116]
House dust mite, ragweed, and Aspergillus fumigatus extracts mixture (DRA)-induced chronic asthma Inducing agent: Sensitization and challenge–intranasal instillation of DRA solution	✓ Persistent peribronchial eosinophilic infiltration ✓ Airways hyperresponsiveness ✓ Smooth muscle cells proliferation ✓ Peribronchial collagen deposition	Chronic asthmatic inflammation, resistant to therapy with cytokine antibodies, thus closely mimicking human chronic asthma	+Presence of human chronic asthma characteristics, such as airway hyperresponsiveness and resistance to cytokine antibodies therapy +Lack of developed tolerance to the inducing allergens −Long duration of the model −Dependence on the purity and concentration of extracts mixture, result in reproducibility issues	[117,118]

+ indicates the advantages of the model; − indicates the disadvantages of the model.

## Data Availability

Not applicable.

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
