# Peer review of "Bronchial Asthma, Airway Remodeling and Lung Fibrosis as Successive Steps of One Process"

_ijms, 2023, doi:10.3390/ijms242216042_

Round 1
Reviewer 1 Report
Comments and Suggestions for Authors
In this study, the authors described the morphological features and molecular mechanisms of asthma-associated airway remodeling and lung fibrosis, as well as the most common murine models and promising biomarkers of asthma progression and post-asthmatic fibrosis.
This is a well-organized review with meaningful results attending the new trends of our knowledge of asthma-fibrosis processes.
It will be helpful for the quality of the article, for the authors to introduce some comments about the role of group-2 innate lymphoid cells in the pathology of asthma-induced fibrosis.
Specifically, they may insert after the line 118 such a comment:
During the last decade, group-2 innate lymphoid cells (ILC2s) have been discovered and successfully established as crucial mediators of lung allergy, airway inflammation, and fibrosis, thus affecting the pathogenesis and clinical course of many respiratory diseases, like for instance asthma, cystic fibrosis, and chronic rhinosinusitis.
[Wirtz S, Schulz-Kuhnt A, Neurath MF, Atreya I. Functional Contribution and Targeted Migration of Group-2 Innate Lymphoid Cells in Inflammatory Lung Diseases: Being at the Right Place at the Right Time. Front Immunol. 2021 Jun 10;12:688879. doi: 10.3389/fimmu.2021.688879. PMID: 34177944; PMCID: PMC8222800.]
Also, they may insert after the line 323 such a comment:
Neutrophil cytosolic factor 1 (Ncf1) is a major genetic factor associated with autoimmune diseases and has been identified as a key player in autoimmune-mediated inflammation. The mice with deficient Ncf1 showed reduced eosinophil infiltration and group 2 innate lymphoid cell (ILC2) activation. We conclude that Ncf1 deficiency enhances Th1 response, deactivates ILC2, and protects against pulmonitis.
[Li M, Zhang W, Zhang J, Li X, Zhang F, Zhu W, Meng L, Holmdahl R, Lu S. Ncf1 Governs Immune Niches in the Lung to Mediate Pulmonary Inflammation in Mice. Front Immunol. 2021 Dec 14;12:783944. doi: 10.3389/fimmu.2021.783944. PMID: 34970267; PMCID: PMC8712564.]
Author Response
Dear Reviewer 1,
We are very grateful to you for your valuable comments and suggestions that helped us to improve the manuscript. We revised and modified the manuscript according to your comments (revised parts are marked by red).
It will be helpful for the quality of the article, for the authors to introduce some comments about the role of group-2 innate lymphoid cells in the pathology of asthma-induced fibrosis.
Specifically, they may insert after the line 118 such a comment:
During the last decade, group-2 innate lymphoid cells (ILC2s) have been discovered and successfully established as crucial mediators of lung allergy, airway inflammation, and fibrosis, thus affecting the pathogenesis and clinical course of many respiratory diseases, like for instance asthma, cystic fibrosis, and chronic rhinosinusitis.
[Wirtz S, Schulz-Kuhnt A, Neurath MF, Atreya I. Functional Contribution and Targeted Migration of Group-2 Innate Lymphoid Cells in Inflammatory Lung Diseases: Being at the Right Place at the Right Time. Front Immunol. 2021 Jun 10;12:688879. doi: 10.3389/fimmu.2021.688879. PMID: 34177944; PMCID: PMC8222800.]
We have added the information regarding group-2 innate lymphoid cells and their role in asthma pathogenesis to the Section 2.1. Please, see lines 158-171.
Also, they may insert after the line 323 such a comment:
Neutrophil cytosolic factor 1 (Ncf1) is a major genetic factor associated with autoimmune diseases and has been identified as a key player in autoimmune-mediated inflammation. The mice with deficient Ncf1 showed reduced eosinophil infiltration and group 2 innate lymphoid cell (ILC2) activation. We conclude that Ncf1 deficiency enhances Th1 response, deactivates ILC2, and protects against pulmonitis.
[Li M, Zhang W, Zhang J, Li X, Zhang F, Zhu W, Meng L, Holmdahl R, Lu S. Ncf1 Governs Immune Niches in the Lung to Mediate Pulmonary Inflammation in Mice. Front Immunol. 2021 Dec 14;12:783944. doi: 10.3389/fimmu.2021.783944. PMID: 34970267; PMCID: PMC8712564.]
We have included the information regarding neutrophil cytosolic factor 1 (Ncf1) in the manuscript. However, in our opinion it is more fitting to add this data to the Section 5 describing asthma biomarkers, since Ncf1 takes part in allergic inflammation, deficiency of Ncf1 gene ameliorates asthma development, and Ncf1 levels correlate with eosinophil infiltration and asthma severity, which points to the possible role of Ncf1 as a prognostic asthma biomarker. Please, see lines 585-596.

Reviewer 2 Report
Comments and Suggestions for Authors
In their review article "Bronchial asthma, airway remodeling and lung fibrosis as successive steps of one process" the authors provide a very broad summary of asthma processes, pathways, and mouse models. Overall the manuscript is fairly well written, but does suffer from a lack of focus. By covering so many different topics, there is not much depth in any individual topic. For instance, the discussion of animal models has the most depth, but it only covers the 2 most common mouse models. Therefore the article would be useful for someone entering the asthma field, but would be of limited use to individuals more familiar with asthma.
Specific comments:
-The title seems a bit misleading a bit misleading to me as the authors don't really have much of a discussion on how bronchial asthma, airway remodeling and lung fibrosis are "successive steps of one process"
-For Table 1, it would help to make the components of the "model" section (model name, inducing agent, and challenge) more easily identifiable such as by separating them with a line or putting more space between them
Author Response
Dear Reviewer 2,
We are very grateful to you for your valuable comments and suggestions that helped us to improve the manuscript. We revised and modified the manuscript according to your comments (revised parts are marked by red).
Overall the manuscript is fairly well written, but does suffer from a lack of focus. By covering so many different topics, there is not much depth in any individual topic. For instance, the discussion of animal models has the most depth, but it only covers the 2 most common mouse models. Therefore the article would be useful for someone entering the asthma field, but would be of limited use to individuals more familiar with asthma.
We fully agree that by trying to cover such a wide variety of topics concerning asthmatic inflammation and airway remodeling we have not delved deep into any of them. However, during our own search for information regarding current state of asthma research, we have noted a distinct lack of contemporary review articles, focused on general description of asthmatic pathology and related processes. For example, most articles on the first two pages of Google Scholar while searching for the phrase “Asthma and airway remodeling” were published between 1999 and 2007, with only one out of 20 published in 2020! It is our hope that, as you have pointed out, our review article would be a good overviewing starting point for a wide range of researchers, who are interested in the topic of asthmatic inflammation, airway remodeling and lung fibrosis, pointing them to the review and research articles, focusing in depth on different aspects of asthma development.
Specific comments:
-The title seems a bit misleading a bit misleading to me as the authors don't really have much of a discussion on how bronchial asthma, airway remodeling and lung fibrosis are "successive steps of one process"
In our manuscript, we tried to reflect the main stages of bronchial asthma development and progression, starting from airways inflammation and ending with irreversible structural changes in the respiratory tract and, in some cases, the lung parenchyma.
We overviewed the main morphological characteristics of bronchial asthma, in particular the development of inflammatory changes in the airway walls, accompanied by alterations in bronchial epithelium, smooth muscle and goblet cells hyperplasia. Next, we reviewed the components of asthma-associated airway remodeling with a focus on subepithelial fibrosis, which in some cases can spread from the airways to the lung parenchyma. Fibrosis of alveolar structures is an important phenomenon, as it usually occurs in the pathogenesis of fatal and long-term asthma and may be associated with disease severity and resistance to therapy. So we believe that the title truthfully reflects the contents of our manuscript, since asthmatic inflammation and remodeling of the airways together with asthma-associated subepithelial fibrosis indeed can be viewed as successive stages of one process.
-For Table 1, it would help to make the components of the "model" section (model name, inducing agent, and challenge) more easily identifiable such as by separating them with a line or putting more space between them
Corrected. The model section of Table 1 was modified according your recommendation. Please, see Table 1.
Reviewer 3 Report
Comments and Suggestions for Authors
The authors present a very interesting and comprehensive review of airway remodeling and fibrosis in asthma. Many of the factors involved in the pathophysiology of asthma are covered.
However, the authors shoul elaborate a bit more on different asthma phenotype and the endotypes involved in the pathophysilogy of each phenotype (eg. different levels and types of inflammation in the airways- not only eosinophilic and neutrophilic, associated comorbidities that may aggravate the underlying condition and airway remodeling etc.).
Additionally, the authors should elaborate more on the biology of IgE since it is the hallmark of alleegic asthma- shortly describe the sensitization vs allergic reactoin processes, the efffects of IgE on effector cells (not only basophils and mast cells).
Line 113: ..."scarification tests.." the authors should clarify what they meant by scarification tests- perhaps skin prick test?
Line 207- "....lead to bronchial subepithelial fibrosis, irreversible structural changes, and fixed airway obstruction." Are the structural changes in asthma really irreversible? The way the authors stated it here makes it sound like they are always irreversible, which is not true. I suggest at least adding "in some cases irreversible.."
Comments on the Quality of English LanguageMinor edits needed
Author Response
Dear Reviewer 3,
We are very grateful to you for your valuable comments and suggestions that helped us to improve the manuscript. We revised and modified the manuscript according to your comments (revised parts are marked by red).
However, the authors shoul elaborate a bit more on different asthma phenotype and the endotypes involved in the pathophysilogy of each phenotype (eg. different levels and types of inflammation in the airways- not only eosinophilic and neutrophilic, associated comorbidities that may aggravate the underlying condition and airway remodeling etc.).
In accordance with your recommendations, we have added an additional section with more thorough description of asthma phenotypes and endotypes. Please, see Section 2.1, lines 71, 83-110.
Additionally, the authors should elaborate more on the biology of IgE since it is the hallmark of alleegic asthma- shortly describe the sensitization vs allergic reactoin processes, the efffects of IgE on effector cells (not only basophils and mast cells).
We have added the paragraph, describing the mechanisms by which IgE facilitates allergen sensitization and asthmatic inflammation development. Please, see lines 126-142.
Line 113: ..."scarification tests.." the authors should clarify what they meant by scarification tests- perhaps skin prick test?
Indeed, by the term “scarification tests” we meant skin prick tests. In order to avoid misunderstanding, we have replaced the term “scarification tests” with “skin prick tests”. Please, see line 152.
Line 207- "....lead to bronchial subepithelial fibrosis, irreversible structural changes, and fixed airway obstruction." Are the structural changes in asthma really irreversible? The way the authors stated it here makes it sound like they are always irreversible, which is not true. I suggest at least adding "in some cases irreversible.."
We fully agree that structural changes in the lungs (remodeling of both airways and lung architecture) in asthma are not always irreversible. Generally, at the initial stages of the disease, morphological changes in the airways and lung tissue are potentially reversible. However, with a long-term, steadily progressing course of asthma with frequent exacerbations and poor response to therapy, structural changes in the respiratory tract and lungs, including fibrotic changes, can become irreversible. In accordance with your recommendations, we have made our statements regarding the irreversible structural changes during asthma progression less drastic throughout the manuscript. Please, see lines 54, 260-262, 426, 538, 608.
Comments on the Quality of English Language
Minor edits needed
According to your recommendation regarding English language and style, the manuscript was copyedited with the help of an AI-based English editing and proofreading web-service QuillBot (https://quillbot.com/grammar-check).

Round 2
Reviewer 3 Report
Comments and Suggestions for Authors
The authors have implemented all required suggestions and changes into the revised version of the manuscript. The revised version is now much clearer and provides an even deeper insight into.the pathophysiology of airway remodeling and fibrosis in asthma.
Author Response
Dear Reviewer 3,
We are very grateful to you for high appraisal of our manuscript!